

# Structure-based identification of potent fibroblast growth factor receptor 4 (FGFR4) inhibitors as potential therapeutics for hepatocellular carcinoma

Lin Fan[1], Hui Xie[1], Weiyu Wang[1], Guizhu Peng[1], Zhen Fu[1] and Qifa Ye[1,2]

[1] Zhongnan Hospital of Wuhan University, Institute of Hepatobiliary Diseases of Wuhan University, Transplant Center of Wuhan University, National Quality Control Center for Donated Organ Procurement, Hubei Key Laboratory of Medical Technology on Transplantation, Wuhan, Hubei, China

[2] The 3rd Xiangya Hospital of Central South University, NHC Key Laboratory of Translational Research on Transplantation Medicine, Changsha, China

Corresponding authors
Zhen Fu, 414271691@qq.com
Qifa Ye, yqf_china@163.com

## ABSTRACT

Fibroblast growth factor receptor 4 (FGFR4), a member of the fibroblast growth factor receptor family, plays a crucial role in cell growth, differentiation, and tissue repair. Increased FGFR4 expression has been detected in various cancers, including lung, liver, kidney and pancreatic cancer, making it a potential drug target. In this study, we conducted a structure-based virtual screening campaign to identify potential FGFR4 inhibitors. The retained compounds were further filtered based on pan assay interference compounds (PAINS) and absorption, distribution, metabolism, excretion, and toxicity (ADME/T) properties, leading to the identification of two promising candidates: MFCD00832235 and MFCD00204244. Quantum mechanical (QM) calculations revealed a large Highest Occupied Molecular Orbital (HOMO) and the Lowest Unoccupied Molecular Orbital (LUMO) (HUMO-LUMO) gaps for both compounds, indicating high dynamic stability and low chemical reactivity. Moreover, the stability of MFCD00832235 and MFCD00204244 at the adenosine triphosphate (ATP)-binding site of FGFR4 was confirmed through molecular dynamics (MD) simulations. The molecular mechanics Poisson-Boltzmann surface area (MM/PBSA) approach predicted favorable binding free energy values for both compounds with the target protein. *In vitro* assay revealed that MFCD00832235 and MFCD00204244 inhibited the growth of HepG2 cells with $IC_{50}$ values of $47.42 \pm 12.93\,\mu M$ and $77.83 \pm 19.17\,\mu M$, respectively. Overall, this study suggested that MFCD00832235 and MFCD00204244 were potential FGFR4 inhibitors and may serve as start points for developing novel modulators of FGFR4 for cancer treatment, particularly hepatocellular carcinoma.

# INTRODUCTION

Fibroblast growth factor receptor 4 (FGFR4), a member of the fibroblast growth factor receptor family, is crucial for regulating cell growth, differentiation, and tissue repair (*Haugsten et al., 2010*). The *fgfr4* gene, located on human chromosome 5q35.1, spans approximately 11.3 kb and consists of 18 exons (*Vainikka et al., 1992*). FGFR4 is essential for epithelial-mesenchymal interactions, which are crucial for organ development and tissue regeneration (*Peláez-García et al., 2013*). By regulating these interactions, FGFR4 helps maintain the structural and functional integrity of tissues (*Khosravi et al., 2021*). It is expressed in liver and plays a key role in bile acid metabolism and liver function regulation (*Raja et al., 2019*; *Shin & Osborne, 2009*). FGFR4 is also significantly expressed in lungs, where it participates in regulating lung development and physiological functions (*Weinstein et al., 1998*), as well in other tissues, including brain, spinal cord, pancreas, and lymph nodes (*Levine et al., 2020*).

Physiologically, FGFR4 is involved in various biological processes such as wound healing, angiogenesis, embryonic development, cell proliferation, and differentiation (*Xie et al., 2020*). However, increased levels of FGFR4 expression have been detected in various cancers, such as breast cancer, hepatocellular carcinoma, renal cell carcinoma, lung cancer, and pancreatic cancer (*Liu et al., 2020*). In hepatocellular carcinoma (HCC), abnormal expression and activation of FGFR4 are closely related to tumor proliferation, invasion and metastasis (*Ho et al., 2009*). Studies show that aberrant elevation of FGFR4 expression is found in 30% of diagnosed HCC patients, which functions as an oncogenic driver pathway (*Oh et al., 2024*). The overexpression of FGFR4 promotes the growth and survival of HCC cells, while inhibiting cell apoptosis.

Fibroblast growth factors (FGFs) are known endogenous ligands of FGFRs. FGF19, in particular, plays a pivotal role in regulating bile acid (BA) synthesis through a negative feedback mechanism. This regulatory process involves postprandial crosstalk between the bile acid-activated ileal farnesoid X receptor (FXR) and the hepatic Klotho beta (KLB) co-receptor, which complexes with FGFR4 to activate downstream signaling pathways (*Li et al., 2024*). FGFR4, a member of the FGFR family, consists of a large extracellular ligand-binding domain composed of three immunoglobulin (Ig)-like subunits (Ig I, Ig II, and Ig III), a transmembrane domain, and two intracellular tyrosine kinase domains. The binding affinity between FGF19 and FGFR4 is enhanced when FGFR4 forms a complex with KLB (*Subbiah & Pal, 2019*). Upon binding to FGF19, the FGFR4–KLB complex activates FGFR4, which can then undergo homo- or heterodimerization and initiate several downstream signaling pathways, including the Ras/Raf and PI3K/AKT pathways (Fig. 1) (*Liu et al., 2020*; *Liu et al., 2021*).

Virtual screening (VS) is a crucial technology in computer-aided drug design, widely employed throughout drug discovery (*Lin, Li & Lin, 2020*). By simulating the interaction between compounds and target molecules, it efficiently identifies potential drug candidates from large number of compound libraries, saving time and resources (*Sadybekov & Katritch, 2023*). Its high-throughput and efficiency enable the processing of millions of compounds rapidly, making it faster and cost-effective than traditional high-throughput

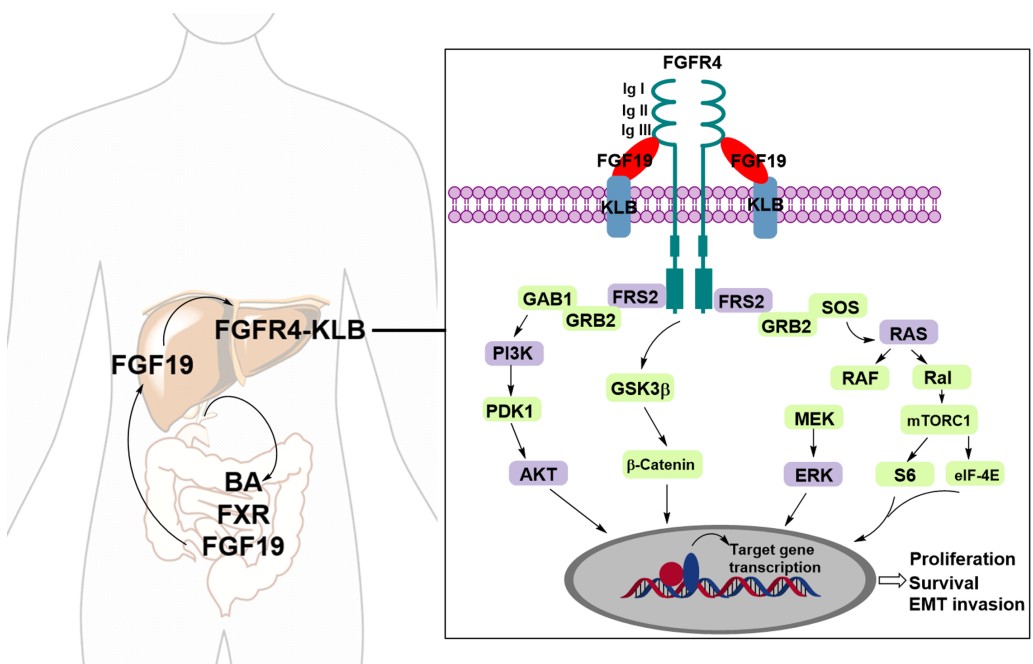

**Figure 1** **The FGF19-FGFR4-KLB signaling pathway.** Upon binding of FGFR4 and Klotho (KLB) to FGF19, the activated FGFR4 undergoes dimerization, either homodimeric or heterodimeric, triggering various downstream signaling pathways.

screening (HTS) (*Szymański, Markowicz & Mikiciuk-Olasik, 2012*). Moreover, virtual screening relies on computer methods, reducing the need for extensive experimental materials and equipment (*Liu et al., 2017*). Using technologies such as molecular docking, pharmacophore modeling, molecular dynamics simulation and QSAR analysis, virtual screening can identify biologically active compounds, optimize their structure and function, enhance activity and selectivity, and accelerate drug development (*Moussa, Hassan & Gharaghani, 2021*).

Here, we conducted a structure-based virtual screening of 53,170 molecules from the Maybridge database to identify potent FGFR inhibitors. The library was initially screened using a cascade docking approach, including LibDock and CDOCKER protocols implemented in Discovery studio 2017 (DS2017). The top 100 scored molecules were clustered and selected based on visual inspection, and the retained compounds were further filtered by pan assay interference compounds (PAINS) and absorption, distribution, metabolism, excretion, and toxicity (ADME/T) properties. Quantum mechanics (QM) calculations were performed to optimize the ligands, followed by all-atom molecular dynamics simulations to gain further insights into the stability and mechanisms of action of the identified hits in complex with FGFR4. Finally, the anticancer activity of the identified hit compounds were evaluated.

## MATERIALS AND METHODS

### Preparation of the receptor file

The X-ray crystal structure of human FGFR4-BLU9931 complex (PDB ID: 4XCU) was taken from the RCSB Protein Data Bank (https://www.rcsb.org/). The structure was prepared by *Protein Prepare tool* embedded in Discovery Studio 2017 (DS2017; Accelrys Software Inc., San Diego, CA, USA). Water molecules were removed, hydrogen atoms were added and missing loop regions were inserted. The protein was protonated at pH 7.4 and minimized under CHARMm filed. The cocrystallized ligand was used to define the centers of the docking site. The coordinates of the center of the docking site were defined as $-13.908$, $4.882$, $10.360$ (x, y, z), and the radius of the box was 8 Å.

### Preparation of screening database

The database of the ligand molecules was obtained from Maybridge containing 53,170 molecules. All molecules were prepared using the Prepare Ligands Tool in Discovery Studio 2017 (DS2017) to generate three-dimensional (3D) structures. Redundant molecules were removed, and the ionization state was determined using a pH-based method, with a minimum pH of 6.5 and a maximum pH of 8.5. The maximum number of tautomers was set to 10, the generation of isomers was enabled, and all other parameters were maintained at their default settings. Finally, a unique 3D conformation for each molecule was generated.

### Virtual screening protocol

LibDock module of DS 2017 was employed for the first-round virtual screening. Docking preference defined as *User specified* with following parameters: *Max Hits to save = 1, Max Humber of hits = 100, Minimum LibDockScore = 100;* Ligand generation method was defined as *Best*. The Smart Minimizer algorithm was performed for *in situ* minimization after docking. All other parameters were set as default. For each molecule, only the best docking pose with highest *LibDockScore* was saved. All output hits were ranked according to the *LibDockScore* and top 2000 of them were retained for further screening.

CDOCKER of the DS2017 was used to perform the second-round docking with high precision. The number of top hits was set to 1 and 10 conformations for each inhibitor were generated with pose cluster radius of 0.5 Å. All the other parameters were set as default. Finally, for each ligand, only one top docking pose was saved. The output ligands were ranked based on the—*CDOCKER Interaction Energy*, and top 100 compounds retained and clustered into 10 clusters. For each cluster, 2 or 3 compounds were selected. This led to the selection of 25 compounds for further analysis.

### Pan assay interference compounds evaluation and ADME/T prediction

The PAINS and ADME/T properties of the 25 active compounds were theoretically investigated by using Swiss-ADME (http://www.swissadme.ch/) and pkCSM (http://biosig.unimelb.edu.au/pkcsm).

## Quantum mechanics calculations

Density functional theory (DFT) is a widely used computational method in scientific research with numerous applications in theoretical studies. In this project, DFT calculations were performed by using DMol3. Geometric optimizations were conducted based on the Local-density approximation (LDA) with the Perdew–Wang function (PWC). Calculated quality was defined as fine, and other parameters were set as default. Following geometric optimization, the orbital properties were calculated. The energies of the Highest Occupied Molecular Orbital (HOMO) and the Lowest Unoccupied Molecular Orbital (LUMO), along with the electron density distributions of these frontier orbitals, were determined using DMol3. The energy gap between HOMO and LUMO has been correlated with the molecular reactivity and further extrapolated to evaluate the activity of the bound inhibitor within the enzyme's catalytic cavity.

## Molecular dynamics (MD) simulation

The FGFR4 complexes with the identified compounds were subjected to all-atom MD simulations using AMBER 2022. The force fields of ff14SB and the General Amber Force Field2 (GAFF2) were used for modeling protein and small molecules, respectively. Counter ions including sodium ($Na^+$) and chloride ($Cl^-$) ions were added to the system for neutralization. The system was solvated in a cubical box of a TIP3P water model and maintained a distance of 12 Å between the solutes and the side of the box. The structure was first minimized by 5,000 steps of steepest descent and 5,000 steps of conjugate gradient. In the process of thermalization, the systems were gradually heated up to 300 K at constant volume over a 500 ps MD simulation. After the thermalization process, a one ns NVT simulation and one ns NPT simulation were performed subsequently. Finally, the system was subjected to a 100 ns simulation with a time step of two fs. The temperature and pressure were maintained at 300 K and one atm using the Berendsen thermostat. Snapshots were saved every 10 ps, yielding 10,000 conformations.

## MD trajectory analysis and MM/PBSA binding energy analysis

To assess the overall molecular system stability, the calculation of root-mean-square deviations (RMSDs) of backbone atoms and ligands, fluctuations of $C\alpha$ atoms (root mean square fluctuations, RMSFs), the hydrogen bonds analysis, the distances between ligands and residues of the binding site were performed by CPPTRAJ program in Amber tools. *MMPBSA.py* script in AMBER 2022 was used to calculate the binding free energies ($\Delta G_{bind}$) for all protein—ligand complexes using molecular mechanics Poisson–Boltzmann surface area (MM/PBSA) method. A hundred snapshots from the last 20 ns equilibration region of trajectory were extracted every 50 ps for the calculation. Entropy contribution was not considered as its calculation might introduce additional error.

## Cell culture and cytotoxicity evaluation

The HepG2 cell line was acquired from the American Type Culture Collection (ATCC). RPMI 1640, Dulbecco's Modified Eagle's Medium (DMEM), and fetal bovine serum (FBS) were obtained from Life Technologies. Cells were initially cultured to approximately 80% confluence in RPMI 1640 or DMEM, supplemented with 10% FBS and 1%

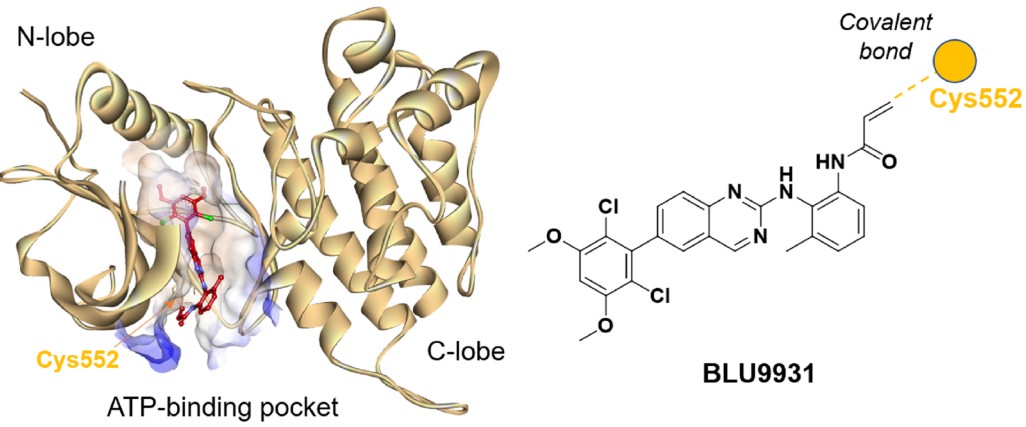

**Figure 2 Crystal structure of BLU9931 in complex with FGFR4 (PDB: 4XCU, left) and the chemical structure of BLU9931 (right).** Cysteine 552 within the FGFR4 hinge region allows for selective inhibition *via* covalent modification.

penicillin/streptomycin, in T25 cell culture flasks at 37 °C in a 5% $CO_2$ atmosphere. For cytotoxicity assessment, cells were seeded into 96-well plates at a density of 2,000 cells per well and incubated for 24 h. Subsequently, the culture medium was replaced, and 100 μL of fresh medium containing varying concentrations of the compounds (0.01, 0.1, 1, 10, 100, 1,000 and 5,000 μM) was added to each well. After 72 h of incubation, the medium was aspirated, and 100 μL of fresh culture medium containing 10 μL of CCK-8 solution (Biosharp) was added to each well. The plate was incubated for an additional 1 h at 37 °C. Optical absorbance was then measured at 450 nm using a microplate reader (BioTek).

## RESULTS AND DISCUSSION

### Analysis of the FGFR4

Understanding the detailed structure of FGFR4, including its secondary domains, adenosine triphosphate (ATP)-binding pocket, and the its interactions with selective inhibitors such as BLU9931, is essential for advancing targeted drug design (Fig. 2). The intracellular region of FGFR4 contains the tyrosine kinase domain (FGFR4-KD), which is critical for its enzymatic activity. This kinase domain is divided into the N-terminal and C-terminal lobes, connected by a regulatory activation segment. The N-terminal lobe comprises mainly β-sheets and an α-helix, while the C-terminal lobe is predominantly α-helical.

BLU9931 selectively inhibits FGFR4 by covalently modifying the unique cysteine residue (Cys552) in its hinge region, which is absent in FGFR1, 2, and 3 where a tyrosine residue is present. In the crystal structure (PDB: 4XCU), BLU9931 forms a covalent adduct with the sulfur atom of Cys552, with hydrogen bonds further stabilizing its interaction with the FGFR4 kinase hinge region. These insights are critical for developing strategies to modulating FGFR4 activity in disease contexts, particularly cancer, where targeted inhibition may offer significant clinical benefits.

## Virtual screening

Virtual screening is an efficient method for rapidly identifying novel drugs from large compound libraries. In this study, cascade virtual screening strategy was employed to identify hit compounds.

A library of 53,170 compounds from Maybridge was screened using a two-round virtual screening process. In the first round, compounds were docked into the binding site of the co-crystal ligand with LibDock for rapid, low-cost screening. The second round utilized CDOCKER for further refinement. For each compound, the conformation with the highest docking score was saved, and the compounds were ranked by docking scores. The top 2,000 compounds with LibDockScore ranged from 122.567 kcal/mol to 185.056 kcal/mol were retained. The retained compounds were subsequently docked into the same binding pocket of FGFR4 with CDOCKER program, an accurate molecular docking technology. As a result, a total of 1,951 poses were obtained after docking. Top 100 compounds with scores above 54.375 kcal/mol were retained on the basis of the—CDOCKER_INTERACTIO_ENERGY scoring function. The retained 100 compounds were clustered into 10 clusters based on FCFP6 finger print, and two or three compounds were selected from each cluster *via* visual inspection. Finally, 25 compounds with diverse scaffolds were selected (Table 1).

## PAINS and ADME/T properties

PAINS are used to identify and exclude compounds that are likely to yield false positives in biological screening assays due to non-specific interference, which can lead to misleading data and resource waste in drug discovery. Therefore, PIANS filtering was performed to the 25 retained compounds using Swiss ADME/T. Results demonstrated that there was no PAINS alert related to the investigated molecules.

Toxicity analysis is a crucial step in drug design, traditionally conducted using *in vivo* animal models, which are time-consuming, costly, and ethically contentious. To overcome these challenges, we predicted the toxicity parameters (genotoxicity, hERG I/II inhibition, hepatotoxicity) of the 25 molecules by employing pkCSM, providing a faster, cost-effective, and ethical alternative. Two compounds, namely MFCD00832235 and MFCD00204244 show no toxic alerts (Table 2). Subsequently, ADME (absorption, distribution, metabolism and excretion) properties of the two compounds were calculated using SWISS web. Both compounds exhibited acceptable pharmacokinetics profiles (Table 3) and were selected for further evaluation.

## Quantum mechanical calculations

Quantum-based geometry optimization determines the most stable molecular configuration by minimizing energy. This method refines initial geometric approximations to achieve greater precision, with the lowest-energy geometry considered the most stable, as molecules naturally tend to minimize their energy spontaneously. Using DMol3, the lowest-energy molecular geometries with fine quality was identified. The 2D structures and 3D optimized geometries of compounds MFCD00832235 and MFCD00204244 were depicted in Table 4.

The concept of frontier molecular orbitals (FMO) is widely used in organic chemistry to analyze molecular structure and reactivity by examining the energy gap between HOMO

**Table 1  Dock scores of the selected 25 compounds.**

| Maybridge ID | Formula | Structure | Docking score (kcal/mol) | |
| --- | --- | --- | --- | --- |
| | | | **LibDock** | **-CDOCKER** |
| MFCD00204244 | $C_{27}H_{32}O_{14}$ | | 147.571 | 67.8585 |
| MFCD00275230 | $C_{26}H_{32}N_2O_{16}S$ | | 154.677 | 65.9605 |
| MFCD03406952 | $C_{25}H_{24}Cl_2N_6O_5$ | | 133.785 | 62.7229 |
| MFCD00220623 | $C_{35}H_{47}N_3O$ | | 135.101 | 60.0623 |
| MFCD00276666 | $C_{33}H_{32}O_{16}$ | | 169.704 | 59.4024 |
| MFCD03086466 | $C_{26}H_{24}Cl_2N_2O_6S$ | | 122.645 | 59.1795 |
| MFCD00277267 | $C_{38}H_{30}O_8S_2$ | | 143.863 | 59.0545 |

**Table 1** (*continued*)

| Maybridge ID | Formula | Structure | Docking score (kcal/mol) | |
|---|---|---|---|---|
| | | | **LibDock** | **-CDOCKER** |
| **MFCD01314044** | $C_{27}H_{24}F_3N_3O_5S$ |  | 127.804 | 59.0397 |
| **MFCD01934358** | $C_{29}H_{42}N_2O_8S_2$ |  | 140.355 | 57.7528 |
| **MFCD05661909** | $C_{26}H_{30}ClN_5O_3$ |  | 141.72 | 57.6059 |
| **MFCD00100989** | $C_{29}H_{48}O_4$ |  | 146.196 | 57.5128 |
| **MFCD00831980** | $C_{25}H_{34}N_4O_8S$ |  | 140.853 | 57.3581 |
| **MFCD00831384** | $C_{26}H_{34}N_2O_{12}S$ |  | 138.493 | 56.8102 |
| **MFCD00832235** | $C_{22}H_{28}Br_2O_6$ |  | 127.077 | 56.5544 |

**Table 1** (*continued*)

| Maybridge ID | Formula | Structure | Docking score (kcal/mol) | |
|---|---|---|---|---|
| | | | **LibDock** | **-CDOCKER** |
| MFCD04123423 | $C_{29}H_{24}FN_3O_5S$ | | 148.706 | 55.6476 |
| MFCD01313115 | $C_{28}H_{35}N_7O_3$ | | 144.868 | 55.6023 |
| MFCD04123788 | $C_{22}H_{24}N_6O_5S_2$ | | 128.298 | 55.4144 |
| MFCD00278503 | $C_{28}H_{36}N_2O_8$ | | 151.073 | 55.3875 |
| MFCD00225611 | $C_{29}H_{28}N_4O_2$ | | 132.626 | 55.3844 |
| MFCD00205402 | $C_{24}H_{32}N_4O_9$ | | 139.53 | 55.1293 |
| MFCD00218330 | $C_{29}H_{40}N_4O_9S_2$ | | 127.153 | 54.9606 |
| MFCD01314205 | $C_{28}H_{35}NO_8S_2$ | | 132.479 | 54.8479 |

**Table 1** (*continued*)

| Maybridge ID | Formula | Structure | Docking score (kcal/mol) | |
|---|---|---|---|---|
| | | | **LibDock** | **-CDOCKER** |
| **MFCD00174945** | $C_{30}H_{28}N_2O_8$ |  | 138.685 | 54.8054 |
| **MFCD04123794** | $C_{23}H_{24}N_4O_5S$ |  | 137.515 | 54.5243 |
| **MFCD03056118** | $C_{35}H_{24}ClN_7O_2$ |  | 152.87 | 54.415 |

**Table 2  Toxic properties of MFCD00832235 and MFCD00204244 predicted by pkCSM.**

| Model name | Predicted value | | Unit |
|---|---|---|---|
| | **MFCD00832235** | **MFCD00204244** | |
| AMES toxicity | No | No | Categorical (Yes/No) |
| Max. tolerated dose (human) | 1.187 | 0.366 | Numeric (log mg/kg/day) |
| hERG I inhibitor | No | No | Categorical (Yes/No) |
| hERG II inhibitor | No | No | Categorical (Yes/No) |
| Oral Rat Acute Toxicity (LD50) | 2.773 | 2.67 | Numeric (mol/kg) |
| Oral Rat Chronic Toxicity (LOAEL) | 1.168 | 5.889 | Numeric (log mg/kg_bw/day) |
| Hepatotoxicity | No | No | Categorical (Yes/No) |
| Skin Sensitisation | No | No | Categorical (Yes/No) |
| T.Pyriformis toxicity | 0.589 | 0.285 | Numeric (log ug/L) |
| Minnow toxicity | −1.958 | 5.967 | Numeric (log mM) |

and LUMO. A narrow gap indicates high reactivity and low kinetic stability, while a wide gap suggests low reactivity and high stability, ultimately influencing a molecule's overall energetic stability. Therefore, to assess the chemical reactivity and kinetic stability of the selected compounds, the HOMO, LUMO, and HOMO-LUMO gap energies were calculated and illustrated in Table 5. The calculated FMO energy band gap values for compounds MFCD00832235 and MFCD00204244 were found to be 3.49 eV and 2.92 eV, respectively. These relatively high values suggested that the molecules possess considerable kinetic stability and low chemical reactivity.

**Table 3 ADME properties of MFCD00832235 and MFCD00204244 calculated using pkCSM.**

| Propeties | Model name | Predicted value | |
|---|---|---|---|
| | | MFCD00832235 | MFCD00204244 |
| Absorption | Water solubility[a] | −6.646 | −2.938 |
| | Caco2 permeability[b] | 1.177 | −0.477 |
| | Intestinal absorption (human)[c] | 94.24 | 20.727 |
| | P-glycoprotein substrate[d] | No | Yes |
| Distribution | VDss (human)[e] | 0.024 | 0.347 |
| | BBB permeability[f] | −0.362 | −1.899 |
| | CNS permeability[g] | −3.007 | −4.864 |
| Metabolism | CYP2D6 substrate | No | No |
| Excretion | Total Clearance[h] | 0.899 | 0.522 |
| | Renal OCT2 substrate[i] | No | No |

**Notes.**
[a] Aqueous solubility descriptor (log mol/L).
[b] Caco-2 cell permeability (log Papp in $10^{-6}$ cm/s > 0.09).
[c] Absorption (human, % > 30).
[d] Ability to inhibit the P-glycoprotein.
[e] Volume of distribution (human, log L/kg) (low if < − 0.15 and high if > 0.45).
[f] Readily crosses the blood–brain barrier if logBB > 0.3 and poorly distributed to the brain if logBB < −1).
[g] Compounds with a logPS > −2 are considered to penetrate the central nervous system (CNS), while those logPS < −3 are considered as unable to penetrate the CNS.
[h] Predicted total clearance log(CLtot) given in log(ml/min/kg).
[i] Assessing whether a given molecule is likely to be an OCT2 substrate.

**Table 4 Quantum-based geometry optimized structures of MFCD00832235 and MFCD00204244.**

| Compound ID | 2D structure | Geometry optimized structure (3D) |
|---|---|---|
| MFCD00832235 |  |  |
| MFCD00204244 |  |  |

## Protein–ligands interaction analysis

Studying protein-ligand interactions is essential in drug discovery for identifying potential candidates and understanding their behavior in biological networks. The analysis of interactions between the two hit compounds and the FGFR4 protein using BIOVIA Discovery Studio Visualizer tools revealed a variety of bonding interactions (Fig. 3).

**Table 5** Representing the asymmetric HOMO, LUMO, and HOMO-LUMO gap energy for selected hit compounds.

| Energy | LUMO (a.u.) | Gap (eV) | HOMO (a.u.) |
|---|---|---|---|
| MFCD00832235 |  | |  |
| | −0.058653 (a.u.) | 3.49513 (eV) | −0.187097 (a.u.) |
| MFCD00204244 |  | |  |
| | −0.10365 (a.u.) | 2.91528 (eV) | −0.210785 (a.u.) |

**Figure 3** The binding modes of MFCD00832235 (A) and MFCD00204244 (B) in complex with FGFR4. The ligands are shown as ball and stick images, while the protein are displayed as surface (left side and middle) and graphic (right side).

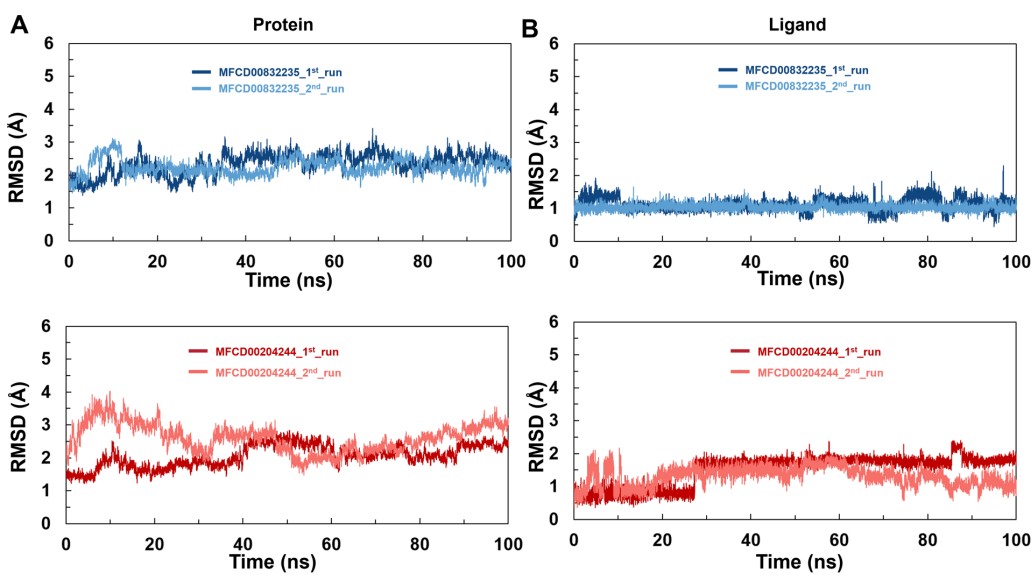

**Figure 4** **Structural fluctuation of the complexes.** (A) Protein backbone RMSD. (B) Ligand RMSD (relative to the protein backbone).

The analysis of the structure of MFCD00832235 in complex with FGFR4 indicated that the compound contacted with the protein mainly through hydrophobic interactions. One of the phenyl rings formed Pi-cation and Pi-sulfur interactions with Lys503 and Met524, respectively. Moreover, Ile534 formed a π-alkyl bond with the same phenyl group. In addition, Leu473 contacted with the Br group at the other phenyl ring (Fig. 3A). In contrast, MFCD00204244 was found to interact with the ATP-binding pocket of FGFR4 primarily through hydrogen bonds. Specifically, five conventional hydrogen bonds were identified in this interaction, including three H-bonds formed between the disaccharide moiety of the ligand and residues Lys503, Glu520 and Asp630 of the protein. Another two H-bonds were attributed to aglycone moiety of the ligand, one formed between the Arg483 and the phenol group, and the other one formed between Asn557 and carbonyl group (Fig. 3B).

## MD simulations

To analyze the behavior of the selected compounds in the macromolecular environment, the hit compounds-bound systems predicted by CDOCKER were subjected to 100 ns MD simulations. The mean root mean square deviation (RMSD) values of protein backbones of MFCD00832235- and MFCD00204244-bound systems were 2.28 Å to 2.33 Å, respectively. It has been found that the simulation was converged after 40 ns for both systems and the RMSD values have been stabilized around a fixed value within the time (Figs. 4A, 4B). Additionally, the mean RMSD values of heavy atoms of MFCD00832235 and MFCD00204244 were 1.07 and 1.40 Å, respectively, indicating excellent initial poses of the complexes.

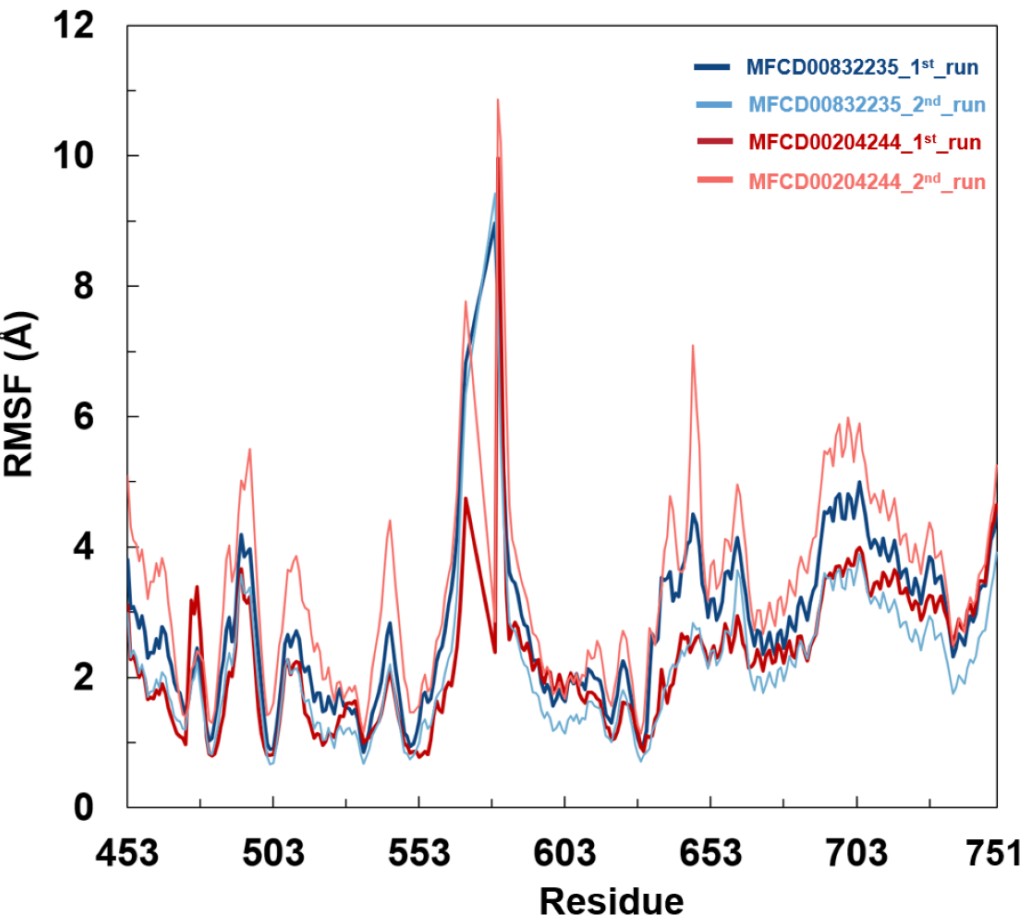

**Figure 5** Root mean square fluctuation (RMSF) of Cα atoms in FGFR4 for MFCD00832235- and MFCD00204244-bound complexes over 100-ns MD simulations. RMSF values were calculated for each residue of FGFR4 in complex with MFCD00832235 (deep and light blue) or MFCD00204244 (deep and light red), across two independent simulation replicates (1st and 2nd run). The x-axis represents residue numbers, and the y-axis shows RMSF values in angstroms (Å). Both complexes exhibit similar RMSF distributions, with moderate fluctuations in the kinase domain and higher flexibility in the C-terminal tail region. Error bars indicate the standard deviation between simulation replicates.

Analyses of root mean square fluctuation (RMSF) *versus* the residue number for studied systems were illustrated in Fig. 5. The protein structures of both systems shared similar RMSF distributions and trends of dynamic features, demonstrated that the binding of MFCD00832235 and MFCD00204244 caused similar backbone conformation changes though they possessed totally different scaffolds. Finally, a timeline was produced to visually represent the conformational variations observed over the 100 ns simulations, showing that the secondary structures of the protein were well aligned and similar zones of fluctuations were observed in consistent with RMSF profiles (Fig. 6).

The stability of MD simulations was further evaluated by measurement of the intermolecular H-bond(s) developed between the FGFR4 complexes of compounds MFCD00832235 and MFCD00204244. Results showed that the complex of MFCD00204244

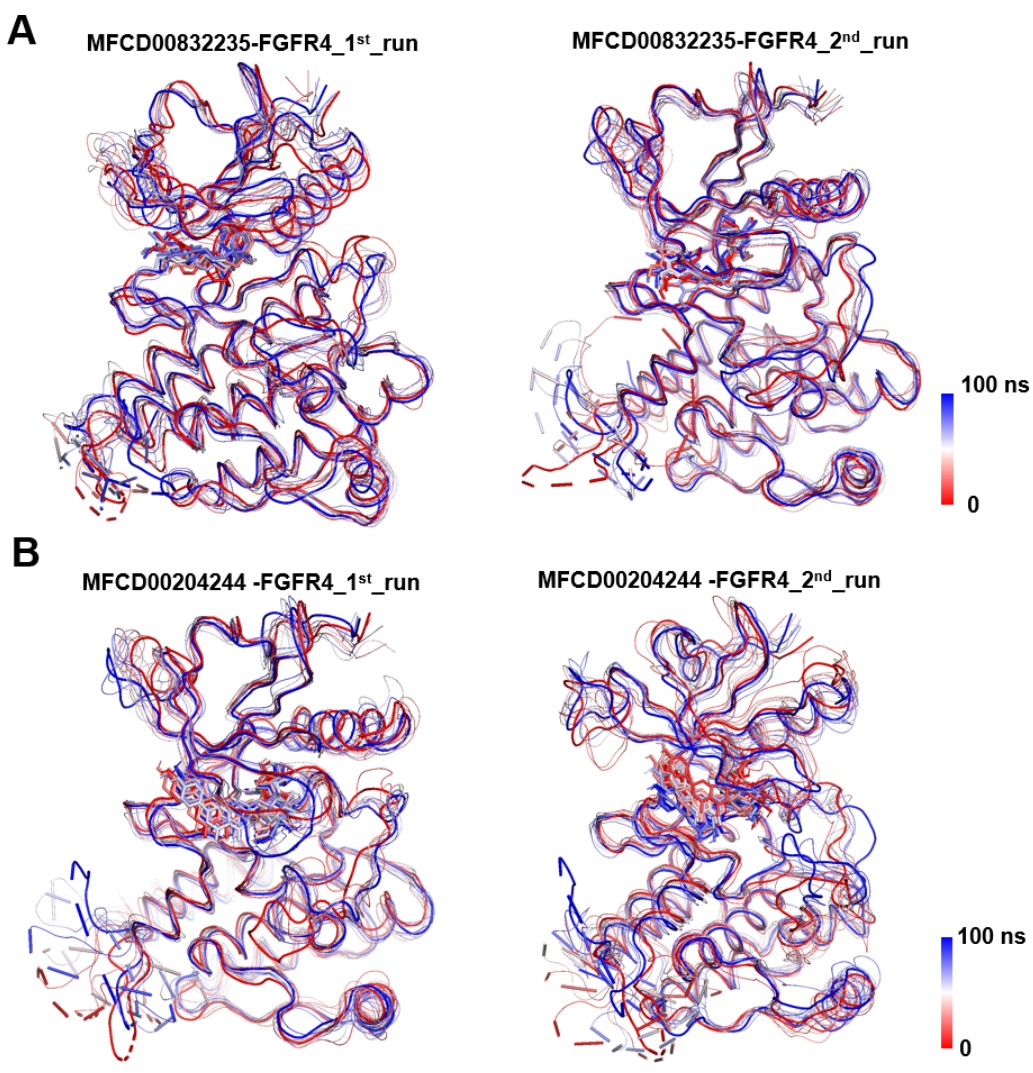

**Figure 6** **Conformational dynamics of FGFR4 complexes with MFCD00832235 and MFCD00204244 over 100-ns MD simulations.** Conformational changes in the MFCD00832235-bound (A) and MFCD00204244-bound (B) systems are visualized using a tube representation, where timestep coloring transitions from red (0 ns) to blue (100 ns) at 10 ns intervals. The initial (0 ns) and final (100 ns) conformations are highlighted with thicker tubes.

formed much more hydrogen bonds throughout the whole 100 ns simulation than that of MFCD00832235 (Fig. 7). This can be explained by the structure of the compound that MFCD00204244 contains more hydroxyl groups in its structure, which greatly facilitates the formation of hydrogen bonds.

A cut-off criterion of $\leq$ 3.5 Å for distance and $\geq$ 120° for the angle between proton donor and acceptor atoms was used to calculate the hydrogen bond (H-bond) occupancy as percentage. Among the hydrogen bonds established by MFCD00832235 during the first run, Asp630 of the main chain contributed to hydrogen bond populating 39.28% and 58.26% of the established hydrogen bonds during the first and second MD simulations,

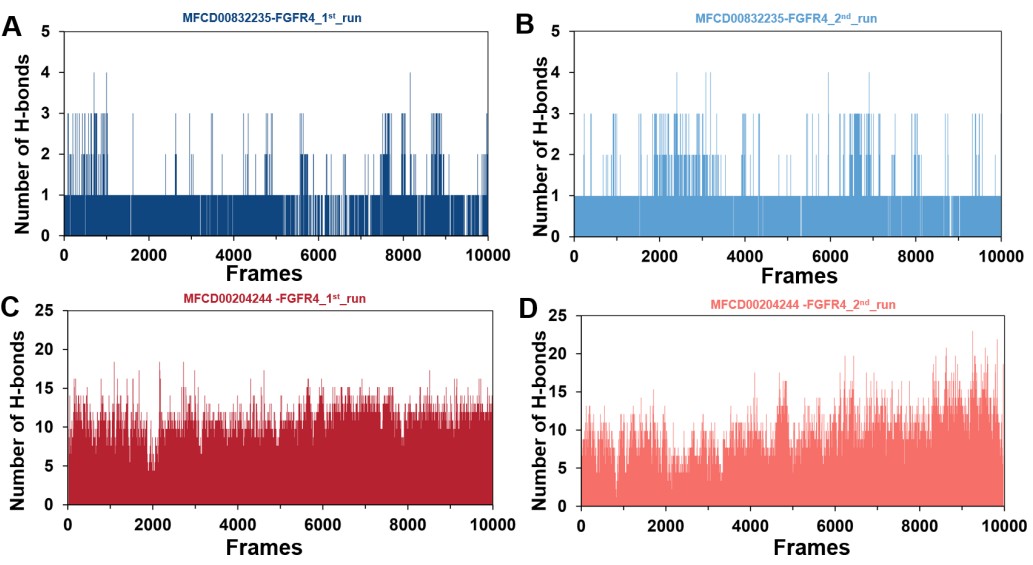

**Figure 7** (A–D) Number of hydrogen bonds established.

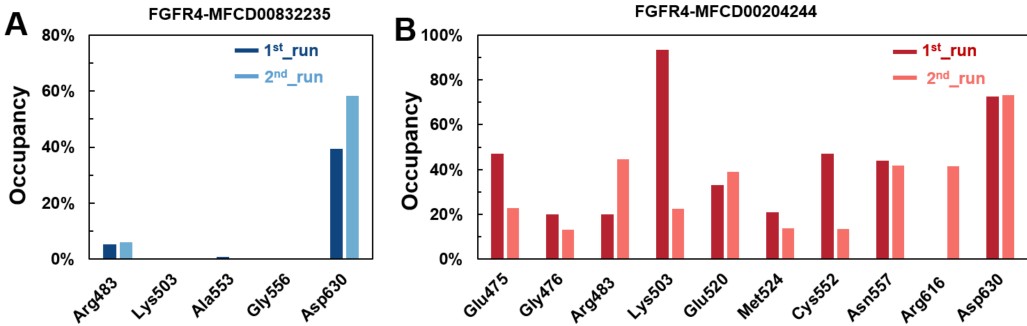

**Figure 8** (A–B) The percentage of occupancy of main residues participating in H-bonding in each system.

respectively. Although the side chain of Arg483 contributed the second most hydrogen bonds populations, it accounted for only 5.30% and 6.09% during the first and second MD simulations, respectively. Consistent with number of hydrogen bonds, MFCD00204244 interacted with more amino acid residues through H-bonds during the two-runs of the MD simulation. Among the hydrogen bonds established by MFCD00204244 during the first run, Lys503 of the side chain established the strongest hydrogen bond populating 93.44%. Meanwhile, Glu520, Asp630 and Asn557 residues contributed to 33.12%, 72.45% and 44.06%, respectively. Similarly, during the second molecular dynamic simulation of MFCD00204244, Asp630 contributed to the most occupancy (73.32%), while Lys503, Glu520 and Asn557 contributed to 43.72%, 39.06% and 41.70% of the established hydrogen bonds, respectively (Fig. 8).

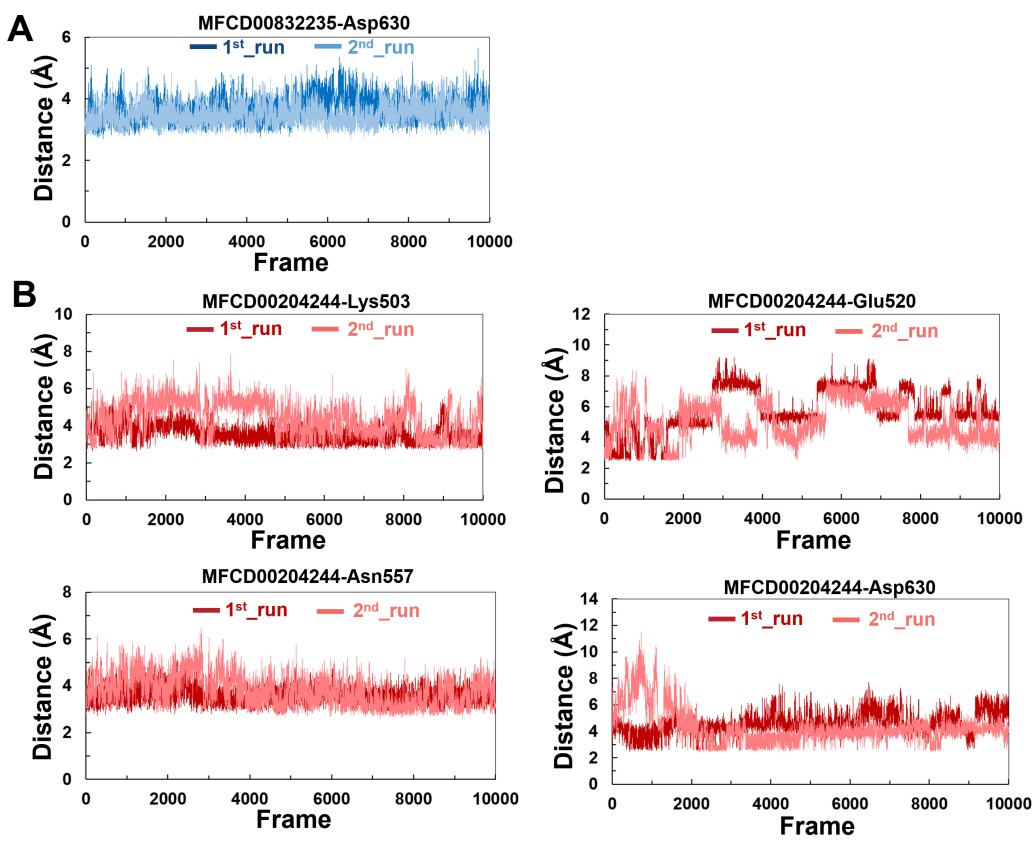

**Figure 9** (A–B) The distance between key binding-site amino acids and the ligands in the frames generated during the 100 ns MD simulations.

The variations in the distances between compounds and amino acid residues involved in hydrogen bonding were evaluated by analyzing the distance between the Cα atoms of the binding cavity and the compounds throughout the MD simulations. In all simulations, the average distances between the heavy atoms of the compounds contributing the hydrogen-bonding residues ranged from 3.5 Å to 6.0 Å (Fig. 9). For the compound MFCD00832235, the distance to Asp630 exhibited slight fluctuations (Fig. 9A). Likewise, the distances to Asn557 and Asp630 showed minor fluctuations as the systems approached stability (Fig. 9B). Conversely, in both MD simulations of MFCD00204244, there were significant fluctuations in the distances to the Glu520 and Lys503 residues (Fig. 9B).

## Binding free energy analysis

Binding affinities of the compounds in complex with FGFR4 were calculated by MM/PBSA approach. Results showed that binding free energies for MFCD00832235 were about two to three folds lower than that of MFCD00204244, with mean values of −33.44 and −14.29 kcal/mol, respectively (Table 6).

Individual components of the binding free energy were also obtained (Table 6). The binding free energy was predominantly influenced by van der Waals interaction and

**Table 6 Energy contribution of the various energetic terms to the total binding free energies of the protein-ligand complexes.**

| Cpd. | Free energy (kcal/mol) | | | | |
|---|---|---|---|---|---|
| | $\Delta E_{ele}$ | $\Delta E_{vdw}$ | $\Delta E_{polar,PB}$ | $\Delta E_{nonpol,PB}$ | $\Delta G_{binding}$ |
| MFCD00832235_1$^{st}$_run | $-9.69 \pm 3.73$ | $-53.31 \pm 3.77$ | $35.95 \pm 4.49$ | $-5.76 \pm 0.18$ | $-32.82 \pm 4.15$ |
| MFCD00832235_2$^{nd}$_run | $-8.04 \pm 2.71$ | $-56.34 \pm 3.23$ | $36.27 \pm 3.94$ | $-5.97 \pm 0.14$ | $-34.06 \pm 3.46$ |
| MFCD00204244_1$^{st}$_run | $-45.78 \pm 8.57$ | $-57.44 \pm 3.55$ | $93.51 \pm 8.54$ | $-7.02 \pm 0.11$ | $-16.72 \pm 6.83$ |
| MFCD00204244_2$^{nd}$_run | $-37.14 \pm 10.81$ | $-55.86 \pm 4.94$ | $88.26 \pm 9.04$ | $-7.12 \pm 0.21$ | $-11.86 \pm 7.29$ |

the nonpolar solvation contribution, which resulted from the burial of the hydrophobic groups of the compounds. The favorable coulomb interactions were counterbalanced by the unfavorable desolvation contributions. In the MFCD00204244-bounded system, the Coulomb interaction contributed—$45.78 \pm 8.57$ kcal/mol (1st_run) and $-37.14 \pm 10.81$ kcal/mol (2nd_run), while the polar solvation component counterbalanced an energy of $93.51 \pm 8.54$ kcal/mol (1st_run) and $88.26 \pm 9.04$ (2nd_run), resulting in a total energy of 47.73 kcal/mol (1st_run) and 51.12 kcal/mol (2nd_run), respectively, which consequently reduced binding affinity. The sum of the two components ($\Delta E_{ele} + \Delta E_{polar,PB}$) in MFCD00832235-bound complex were 26.26 kcal/mol (1st_run) and 28.23 kcal/mol (2nd_run), respectively, much lower than those in MFCD00204244-bounded system. In addition, due to the prohibitively high computational cost and relatively low prediction accuracy associated with NMODE analysis, the entropy change is always neglected.

## Anti-cancer activity

We further evaluated the *in vitro* anticancer activities of MFCD00832235 and MFCD00204244 against HepG2 cells using the CCK8 assay, with 5-fluorouracil (5-FU) as a positive control (Fig. 10). The results demonstrated that MFCD00832235 and MFCD00204244 displayed moderately cytotoxic towards HepG2 cells with an IC$_{50}$ of $47.42 \pm 12.93$ μM and $77.83 \pm 19.17$ μM, respectively. These results indicated that both compounds offered potential candidates for the treatment of liver cancer.

In summary, we have identified two molecules, MFCD00832235 and MFCD00204244 as potential therapeutics for HCC. Although their anti-cancer activity is less potent than the positive control, the activity at μM level is acceptable as hit compounds. Both compounds can be further optimized through structure—activity relationship (SAR) studies to improve their activity. The optimization of MFCD00832235 could be performed in the following aspects. Firstly, the methoxy groups on the benzene rings can be replaced with other groups to explore the effect of different electronic properties on activity. Additionally, the ethylene glycol ether linker between the two benzene rings can be optimized, for example, by introducing amide bonds, amino groups, *etc*. Compound MFCD00204244 is a glycoside compound with high hydrophilicity, which may limit its ability to cross the cell membrane. Therefore, efforts could be made to retain the aglycone while replacing the disaccharide module with other groups, or by introducing hydrophobic groups. In conclusion, this work provides a starting point for the development of drugs for the treatment of hepatocellular carcinoma (HCC).

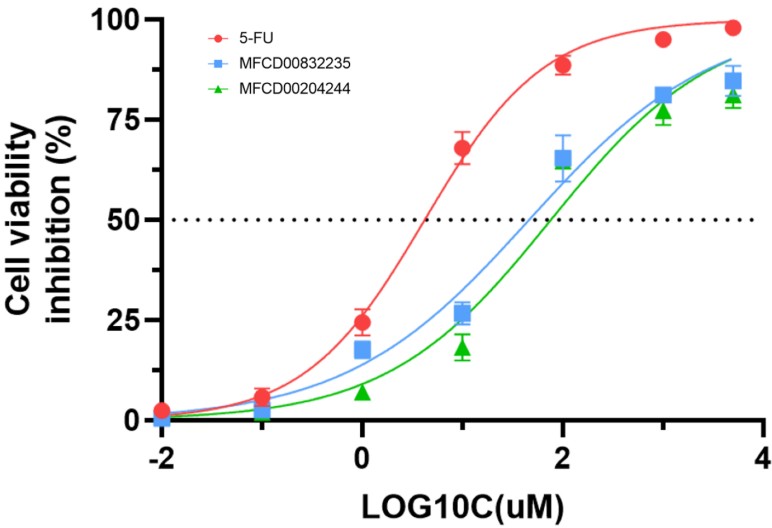

**Figure 10** **Inhibition (%) of cell viability by the two hit compounds and 5-FU as positive control.** IC50 values are the mean of three independent experiments.

## CONCLUSION

This study employed advanced computational methods to identify potential FGFR4 inhibitors. Although there exist several inhibitors target FGFR4, there is no such drug in clinical use yet. To discover novel and effective small molecule FGFR4 inhibitors, we implemented a systematic workflow. First, 53,170 compounds from the Maybridge database were screened by a two-round virtual screening process using Libdock and CDOCKER. The top 2,000 compounds from the first round were further docked using CDOCKER, resulting in 1,951 poses. The top 100 compounds were clustered into 10 groups, and two to three compounds from each group were selected by visual inspection, ultimately choosing 25 compounds with diverse structures for further analysis. To ensure the safety and efficacy of the compounds, PAINS filtering and ADME/T property evaluations were performed. Results indicated that two compounds, MFCD00832235 and MFCD00204244, had no toxicity alerts and demonstrated good pharmacokinetic properties. To investigate and optimize the geometry of the compounds, a computational DFT based QM calculation has been performed, and the calculated HOMO-LUMO gaps indicated considerable kinetic stability and low chemical reactivity. We further investigated the binding modes of the two hits in complex with FGFR4. The 100 ns molecular dynamics simulations showed stable RMSD values and similar backbone conformational changes. Binding affinity calculation using MM/PBSA methods demonstrated that MFCD00832235 exhibited lower binding free energy than MFCD00204244, mainly influenced by van der Waals interactions and nonpolar solvation contributions. Finally, the *in vitro* assay showed that compounds MFCD00832235 and MFCD00204244 inhibited the growth of HepG2 cells with $IC_{50}$ values of $47.42 \pm 12.93$ μM and $77.83 \pm 19.17$ μM, respectively.

Overall, MFCD00832235 and MFCD00204244 demonstrated to be potential FGFR4 inhibitors, particularly for the treatment of hepatocellular carcinoma. However, this study is entirely based on computational screening, and the identified compounds have not yet been experimentally validated *in vitro* or *in vivo*. Therefore, further studies are needed to confirm their inhibitory activity against FGFR4 and their mechanisms of action. Additionally, the virtual screening methods used in this study can be applied to identify potential inhibitors of other cancer-associated proteins, although results may vary depending on the target protein's structure and function and the properties of the screened compound libraries.

### Funding
This work was supported by the Research Supporting project (Grant No. 82200734) by the National Natural Science Foundation of China. The funders had no role in study design, data collection and analysis, decision to publish, or preparation of the manuscript.

### Grant Disclosures
The following grant information was disclosed by the authors:
The Research Supporting project by the National Natural Science Foundation of China: No. 82200734.

### Competing Interests
The authors declare there are no competing interests.

### Author Contributions
- Lin Fan conceived and designed the experiments, performed the experiments, analyzed the data, prepared figures and/or tables, writing-original draft, data analysis, investigation, software, and approved the final draft.
- Hui Xie conceived and designed the experiments, performed the experiments, analyzed the data, prepared figures and/or tables, data curation, visualization, software, and approved the final draft.
- Weiyu Wang conceived and designed the experiments, prepared figures and/or tables, authored or reviewed drafts of the article, writing-review and editing, and approved the final draft.
- Guizhu Peng conceived and designed the experiments, performed the experiments, prepared figures and/or tables, authored or reviewed drafts of the article, writing-review, and approved the final draft.
- Zhen Fu conceived and designed the experiments, performed the experiments, analyzed the data, prepared figures and/or tables, conception, methodology, data validation, and approved the final draft.
- Qifa Ye conceived and designed the experiments, performed the experiments, analyzed the data, prepared figures and/or tables, conception, software, data curation, resources, formal analysis, project administration, and approved the final draft.

## Data Availability

The raw measurements are available in the Supplementary File.

## Supplemental Information

Supplemental information for this article can be found online at http://dx.doi.org/10.7717/peerj.19183#supplemental-information.

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
