# Peer review of "Structure-based identification of potent fibroblast growth factor receptor 4 (FGFR4) inhibitors as potential therapeutics for hepatocellular carcinoma"

_PeerJ, doi:10.7717/peerj.19183_

## Round 0.1 · original submission · Major Revisions

Please address issues pointed by the reviewers and amend manuscript accordingly.

Reviewer 1 ·

Basic reporting

This work is interesting and somewhat meaningful. But the results, or the findings of two compounds are not that of being convinced. I have several questions about this manuscript.
1. The two compounds have very different structure, as MFCD00204244 is full of hydroxy while MFCD00832235 has many -OCH3. Why both of them are highly compatible with FGFR4?
2. what's the change of FGFR4 expression in HCC? what's the results of two compounds binding to FGFR4? are they stabilized or degraded FGFR4?
3. Fig2, why the structure of FGFR4 seems to be not identical in pannels A and B?

Experimental design

no comment

Validity of the findings

no comment

Reviewer 2 ·

Basic reporting

This article is not suitable for publication because
1, FGFR4 studies are not new. Currently, there are clinical studies in phase 2 such as infigratinib, erdafitinib and many medications in this group were terminated after pre-clinical phase trial failed.

Suggestions
1, There should be information on how much FGFR4 with two protein compound there is, because from the report, in the literature that FGFR4 was found 5-10% of all human cancer.
2, It may be useful as a reference for finding other specific proteins compound by computer-based screening.
3, I would like the authors to add up the figure or picture of mechanism of action of FGFR4 in hepatocellular carcinoma.
4, The title of this research should be added ....potential for HCC treatment in the future.

Experimental design

The experimental design and material preparation are very good and meet the intended objectives. However, the results are only suggestive and cannot be concluded that they will be useful in creating new innovations because they have not been tested in the laboratory or in living.

Validity of the findings

Because this study used computers to help analyze and screen, there has been no validation in both the laboratory and living things to see if it is true or not. Therefore, further studies are needed.

Additional comments

1. In fact, the topic is not really relevant to the research because it is a study using computer analysis. There are no studies in vitro or in vivo. It is only a prediction that it will be useful in finding new drugs. However, the results are still far from real life.
2. Currently, we have drugs related to tyrosine kinase pathways such as sorafenib, lenvatinib for treating liver cancer, but they can extend life by less than 15 months. From this study, it was found that 2 protein compounds are potential useful for making new drugs but finding are from computer analysis. This research will be interesting by there should be study in both vitro and vivo, if the results are positive, a pilot study should be done for proving drugs containing both protein compounds are more effective than existing drugs.

Reviewer 3 ·

Basic reporting

The results presented in this paper is clear. Few highly related references have not be cited. The authors need open source code and data.

Experimental design

The methods section need contain more details for the proposed method. Although some of them become popular, it still need present them in more reasonable framework.

Validity of the findings

The findings are only based on computational results. The experiment validations are expected.

Additional comments

In this paper, the authors conducted a computational study to identify potent inhibitors of fibroblast growth factor receptor 4 (FGFR4) for the treatment of hepatocellular carcinoma. The following lists some comments. First, the computational screening was not complemented with experimental validation, which is crucial for confirming the inhibitory activity of the identified two compounds. Second, the study relied on predictive models for ADME/T properties and toxicity, which may not fully capture the in vivo behavior of the compounds. Third, it need provide detailed discussion on the clinical implications and potential pharmacological optimization of the identified inhibitors. Fourth, the sample size of potential inhibitors was limited, which may restrict the generalizability of the findings. Moreover, it is expected to expand the compound library for a more comprehensive screening, and conduct a thorough pharmacological and toxicological evaluation of the candidate inhibitors. The validation of these computational results are very important.

---

## Round 0.2 · accepted · Accept

All issues pointed out by the reviewers were adequately addressed and the revised manuscript is acceptable now.

Reviewer 2 ·

Basic reporting

no comment

Experimental design

no comment

Validity of the findings

no comment

Additional comments

This study on biomarkers for drug development is considered interesting and acceptable according to PeerJ's guidelines. Some wrong words need correction before publication. Additionally, I would accept studies conducted both in vitro and in vivo, not just computer based.

Reviewer 3 ·

Basic reporting

The authors have improved their paper.

Experimental design

The authors provided computational experiments and wet-lab experiment.

Validity of the findings

The authors have validated the findings via an experiment.

Additional comments

The authors need formulate the findings that have been validated by experiments. And those that need more experimental validations can be clarified.